# Potential Misidentification of Natural Isomers and Mass-Analogs of Modified Nucleosides by Liquid Chromatography–Triple Quadrupole Mass Spectrometry

**DOI:** 10.3390/genes13050878

**Published:** 2022-05-13

**Authors:** Xiuying Lin, Qianhui Zhang, Yichao Qin, Qisheng Zhong, Daizhu Lv, Xiaopeng Wu, Pengcheng Fu, Huan Lin

**Affiliations:** 1State Key Laboratory of Marine Resource Utilization in South China Sea, Hainan University, Haikou 570228, China; yingikun9@163.com (X.L.); 18095132210034@hainanu.edu.cn (Y.Q.); pcfu@hainanu.edu.cn (P.F.); 2College of Food Science and Engineering, Hainan University, Haikou 570228, China; zhangqianhui000@163.com; 3Shimadzu (China) Corporation, Guangzhou Branch, Guangzhou 510656, China; skczqs@shimadzu.com.cn; 4Analysis and Testing Center, Chinese Academy of Tropical Agricultural Sciences, Haikou 571101, China; ldz162000@126.com (D.L.); dygxzx@126.com (X.W.)

**Keywords:** nucleoside misidentification, RNA modification, epitranscriptomics, LC-MS/MS, tRNA, HILIC

## Abstract

Triple quadrupole mass spectrometry coupled to liquid chromatography (LC-TQ-MS) can detect and quantify modified nucleosides present in various types of RNA, and is being used increasingly in epitranscriptomics. However, due to the low resolution of TQ-MS and the structural complexity of the many naturally modified nucleosides identified to date (>160), the discrimination of isomers and mass-analogs can be problematic and is often overlooked. This study analyzes 17 nucleoside standards by LC-TQ-MS with separation on three different analytical columns and discusses, with examples, three major causes of analyte misidentification: structural isomers, mass-analogs, and isotopic crosstalk. It is hoped that this overview and practical examples will help to strengthen the accuracy of the identification of modified nucleosides by LC-TQ-MS.

## 1. Introduction

Research into post-transcriptional RNA modification is increasingly focusing on its critical impacts on RNA decay, translational efficiency, subcellular localization, quality control, RNA–protein interactions, and disease development [1,2,3]. Intensely mined information from the epitranscriptome reveals that dynamic modification of RNA is a response to physiological and environmental changes, although the biological consequences of such modifications are still being elucidated [3,4,5]. Methylation of key transcripts, for example, *N*^6^-methyladenosine (m^6^A), has been observed in budding yeast and was crucial to the initiation of meiosis during nitrogen starvation [6]. YTHDF1, an m^6^A reader protein that enhances translational efficiency by recruiting eukaryotic initiation factor 3, was found to be concentrated on stress granules and triggered stalled translation when arsenite was added to induce oxidative stress in HeLa cells [7]. YTHDF2 (an m^6^A reader) and FTO (an m^6^A eraser) competitively bind to the 5′-UTR of messenger (m)RNA and regulate m^6^A methylation in a mouse embryonic fibroblast (MEF) cell line, which facilitates cap-independent translation of specific transcripts under stress conditions [8]. On transfer RNAs (tRNAs), modifications at positions 34 (the wobble position), 37, and 58 can be dynamic and related to environmental factors. When taurine supply was limited, τm^5^U34 (5-taurinomethyluridine) and τm^5^s^2^U34 (5-taurinomethyl-2-thiouridine) on five mitochondrial transfer (mt)RNAs switched to cmnm^5^U34 (5-carboxymethylaminomethyluridine) and cmnm^5^s^2^U (5-carboxymethylaminomethyl-2-thiouridine) with unknown consequences [9]. Bicarbonate-free air culturing of HEK293T cells triggered the downregulation of t^6^A37 (*N*^6^-threonylcarbamoyladenosine) on mt-tRNAs decoding ANN codons and probably impaired the translation of mitochondrial respiratory complex I [10]. m^1^A58 (1-methyladenosine) in human cytoplasmic tRNAs was influenced by the glucose concentration of the culture medium via an FTO (m^1^A eraser)-dependent pathway that adjusted the decoding preference [11].

To determine the identity, quantity, and location of RNA modifications, RNA sequencing (NGS or NNGS approaches) [12,13,14], oligonucleotide mass spectrometry [15,16], and nucleoside mass spectrometry [17] are frequently used. Nuclear magnetic resonance spectroscopy has also recently been used to observe the dynamic incorporation of RNA modifications into nascent tRNA [18,19]. A generic protocol to qualitatively and quantitatively analyze modified nucleosides has been developed using high-performance liquid chromatography coupled to triple quadrupole mass spectrometry (LC-TQ-MS) [20]. Total RNA or a purified fraction is hydrolyzed and dephosphorylated to the free nucleosides using alkaline phosphatase, phosphodiesterase I, and nuclease P1. Two-step digestion is recommended as basic pH is suitable for alkaline phosphatase, while phosphodiesterase I and nuclease P1 are more effective in acidic conditions. However, some studies recommend an acidic environment for the complete digestion because certain RNA modifications are sensitive to pH. For instance, ct^6^A (cyclic *N*^6^-threonylcarbamoyladenosine) quickly epimerizes to t^6^A in mild alkaline buffer, causing an 18 Da mass increase [21]. The enzymes are then removed from the digestion mixture using a 10 kDa cut-off centrifugal filter unit, and the resulting nucleosides are vacuum-dried and dissolved in an appropriate solvent, usually 90% (*v*/*v*) acetonitrile or water, depending on the downstream liquid chromatography. Hydrophilic interaction liquid chromatography (HILIC) [22] and reversed-phase chromatography (RPC) [23] with semi-micro flow rates (0.1–1 mL/min) have been used to separate nucleosides, while micro-flow HPLC (5–50 μL/min) has recently been recommended for higher sensitivity without loss of reproducibility [24].

Multiple reaction monitoring (MRM) mode is used with TQ-MS when determining modified nucleosides. Collision-induced dissociation of the *N*-β-glycosidic bond under the appropriate collision energy (CE) yields the nucleobase production, BH_2_^+^. Ideally, synthetic standards of the modified nucleosides should be used to confirm optimum CE, ion mass–to–charge ratios (*m*/*z*), specific product ions and retention times, but their preparation is expensive, time consuming and labor intensive. The *m*/*z* values for precursor (MH^+^) and product (BH_2_^+^) ions can be readily calculated from their chemical structures (Figure 1). CEs for detecting modified nucleosides can be estimated using native (unmodified) adenosine (A), uridine (U), cytidine (C), and guanosine (G) standards. This may not yield exact values for the modified nucleosides but is a practical compromise. Some published LC-TQ-MS applications for RNA modifications use fewer than 20 modified nucleoside standards and determine other nucleosides using calculated MRM transitions and estimated CE values [20,25].

Over 160 natural RNA modifications have been identified to date [26]. However, this complexity and the low resolution of TQ-MS (approximately 0.5 Da) can give rise to three types of misidentification when using MH^+^ and BH_2_^+^ MRM transitions to determine nucleosides. Type I mistakes result from regioisomers. For instance, five natural monomethylated adenosines have been identified. These are m^1^A, m^2^A (2-methyladenosine), m^6^A, m^8^A (8-methyladenosine), and Am (2′-*O*-methyladenosine) (Figure 2), and have been found variously in mRNA, tRNA, and ribosomal (r)RNA [26]. Am can be discriminated by monitoring the transition *m*/*z* 282.1→136 because of its unmodified nucleobase, while m^1^A, m^2^A, m^6^A, and m^8^A all give rise to the transition *m*/*z* 282.1→150 (Figure 2). High-resolution mass spectrometry can discriminate between these four monomethylated derivatives via in-source, collision-induced dissociation (CID) and negative mode ionization that produces unique patterns of fragments [27]. However, this approach requires a library of MS^2^ and MS^3^ spectra for each isomer and sensitivity can be reduced for quantitative analysis.

Although synthetic standards can strengthen the identification of nucleoside isomers, some cannot be separated (or are closely eluted) by liquid chromatography because of their structural similarity. Type II mistakes with TQ-MS are nucleoside misidentifications due to similar masses (<0.5 Da mass differences). m^6,6^A (*N*^6^,*N*^6^-dimethyladenosine) and f^6^A (*N*^6^-formyladenosine) exhibit MH^+^ masses of 296.1359 and 296.0995, respectively. Both are modified on the nucleobase (Figure 3). Low-resolution TQ-MS of *m*/*z* 296.1→164 cannot distinguish between these two compounds. Furthermore, m^6,6^A has an isomer—m^2,8^A (2,8-dimethyladenosine)—which can lead to the complicated situation of f^6^A/m^6,6^A being present in a eukaryotic total RNA sample, or m^6,6^A/m^2,8^A being present in a eubacterial rRNA sample [26].

Type III misidentification arises from isotopic crosstalk, which is often not considered. For instance, the low-resolution isotopic distributions of positively ionized adenosine and inosine are shown in Figure 4. The transition 269.1→137 used to monitor inosine is subject to interference from the same transition arising from an isotopologue of adenosine. When adenosine and inosine are eluted simultaneously by a short HPLC method, the signal for inosine would be amplified significantly by isotopic mass of adenosine. Given the abundance of isotopic masses of small molecules composed of the elements C, H, O, N, and S, nucleosides differing in mass by one or two units are likely to interfere with each other. Based on the natural abundance of isotopes of these elements (13C 1.11%, 2H 0.0115%, 18O 0.205%, 15N 0.364%, and 34S 4.21%), such mass differences mainly arise from 13C and 34S. Such mass-analogs are not rare among naturally modified nucleosides: crosstalk between mcm^5^U (5-methoxycarbonylmethyluridine, 317.1→185) [28], nchm^5^U (5-carbamoylmethyl-2-thiouridine, 318.1→186) [29], and cm^5^s^2^U (5-carboxymethyl-2-thiouridine, 319.1→187) [30] is a good example and is illustrated in Figure 5.

These three types of misidentifications can occur in the same RNA sample; thus, correctly identifying a nucleoside by TQ-MS is not straightforward. This study illustrates this complexity by describing the analysis of 17 nucleosides and modified nucleosides using three commercially available liquid chromatography columns. Ongoing discussion of the separation of commonly modified nucleosides using reversed-phase and HILIC chromatography, and of the signals derived from TQ-MS, will help to minimize the misidentifications described above.

## 2. Materials and Methods

LC-MS grade formic acid and acetonitrile were purchased from Thermo Fisher (Shanghai, China) and LC-MS grade water from Sigma (Shanghai, China) for the preparation of mobile phases and standard solutions. Nucleoside standards were purchased from respected commercial manufacturers and agencies as follows: uridine (U), cytidine (C), 4-thiouridine (s^4^U), and *N*^6^-methyladenosine (m^6^A) from Aladdin (Shanghai, China); 1-methyladenosine (m^1^A), 1-methylinosine (m^1^I), *N*^6^,*N*^6^-dimethyladenosine (m^6,6^A), *N*^6^-formyladenosine (f^6^A), pseudouridine (Y), 5-hydroxyuridine (ho^5^U), 2-thiouridine (s^2^U), 5-methyldihydrouridine (m^5^D), 2-thiocytidine (s^2^C), and *N*^1^-methylguanosine (m^1^G) from TRC (Toronto, Canada); 2-methyladenosine (m^2^A) from Howei Pharm (Guangzhou, China); *N*^2^-methylguanosine (m^2^G) from TopScience (Rizhao, China); and 7-methylguanosine (m^7^G) from Sigma (Shanghai, China).

Stock solutions of nucleosides (10–100 mM) were prepared in dimethyl sulfoxide and stored at −20 °C. Mixed standard solutions were prepared by diluting stock solutions with 0.1% formic acid in acetonitrile/water (90/10, *v*/*v*) for HILIC chromatography, or water containing 0.1% formic acid for reversed-phase chromatography.

Mass spectrometry was conducted on a QTRAP 6500 LC-MS system (AB Sciex, Redwood, CA, USA). Mobile phase A consisted of 0.1% (*v*/*v*) formic acid in water, and mobile phase B was acetonitrile containing 0.1 % (*v*/*v*) formic acid. Reversed-phase chromatographic separations were run on a Waters Atlantis T3 column (2.1 mm^2^ × 150 mm^2^, 3 µm) and a Supelco Discovery HS F5 column (2.1 mm^2^ × 150 mm^2^, 3 µm). The gradient program (0.1 mL/min flow rate) was as follows: 0–6 min, 0% B; 6–35 min, 0–90% B; 35–40 min, 90% B; 40–40.1 min, 90–0% B; and 40.1–50 min, 0% B. HILIC separations were conducted on a Waters Acquity UPLC BEH amide column (2.1 mm^2^ × 150 mm^2^, 1.7 µm) at 0.1 mL/min flow rate using the elution gradient: 0–5 min, 90% B; 5–35 min, 90–40% B; 35–40 min, 40% B; 40–40.1 min, 40–90% B; and 40.1–50 min, 90% B. Column temperatures were maintained at 36 ºC, the autosampler at room temperature, and the injection volume was 1 μL. MS and MS/MS detection used an electrospray ionization (ESI) source in positive ion mode and the following optimized parameters: ion-spray voltage 5.5 kV, source temperature 350 °C, curtain gas 30 psi, collision gas 8 psi, ion source gas 1 at 30 psi, ion source gas 2 at 1 psi, entrance potential 10 V, collision cell exit potential 13 V, and dwell time 10 ms. MultiQuant 3.0.3 software (AB Sciex, Redwood, CA, USA) was used for data analysis.

## 3. Results and Discussion

### 3.1. Analytical Chromatography Columns

Three analytical columns were evaluated for their ability to resolve modified nucleosides over a 50 min chromatographic runtime: Acquity BEH amide (1.7 μm, 2.1 mm × 150 mm; Waters), Discovery HS F5 (3 μm, 2.1 mm × 150 mm; Supelco), and Atlantis T3 (3 μm, 2.1 mm × 150 mm; Waters) (Figure 6). The Acquity BEH amide (HILIC) column was packed with ethylene-bridged hybrid particles covalently attached by trifunctionally-bonded amide groups, while the linker structure of BEH amide is not published by the Waters Corporation. The Discovery HS F5 column was filled with spherical silica gel and a propyl spacer-linked pentafluorophenyl (PFP) stationary phase. The Atlantis T3 column was an octadecyl silica-based (ODS), reversed-phase C18 column with optimized pore diameter, C18-ligand density, and end-capping. Excellent performance in the separation of highly polar chemicals, including carbohydrates and nucleoside triphosphates, has been reported using these columns [31,32,33].

### 3.2. Adenosine Derivatives

One pmol each of m^1^A, m^2^A, m^6^A, m^1^I (1-methylinosine), m^6,6^A, and f^6^A were mixed and resolved on the three columns (Figure 7). Positive mode mass pair 282.1→150 was monitored to detect m^1^A, m^2^A, and m^6^A, 283.1→151 for m^1^I, and 296.1→164 for m^6,6^A and f^6^A. The methylated adenosines were eluted in the order m^6^A/m^2^A/m^1^A under HILIC conditions, but m^1^A/m^2^A/m^6^A under PFP and ODS conditions (Figure 7A). With the PFP column, m^2^A and m^6^A did not achieve baseline separation, which may lead to type I misidentification.

When *m*/*z* 283.1→150 was used to detect m^1^I, type III misidentification (isotopic crosstalk) could occur. Interfering signals from isotopes of m^1^A, m^2^A, and m^6^A were evident, adjacent to the primary m^1^I peak (Figure 7B). In the case of HILIC separation, a small m^6^A peak eluted close to m^1^I, while a m^2^A peak co-eluted with m^1^I under PFP conditions. Isotopic crosstalk should be carefully ruled out, especially when the target analyte is in low abundance and may be masked by an isotopologue of another nucleoside.

m^6,6^A and f^6^A exhibited similar retention times in the HILIC method (Figure 7C), although baseline separation was achieved. TQ-MS is not sufficiently sensitive to distinguish the mass difference between these two nucleosides (0.0364 Da); thus, the transition 296.1→164 acquired signals from both compounds. It is noteworthy that, although one pmol of each standard was injected, the signal for f^6^A was much lower than for m^6,6^A. This can be explained by the *N*^6^-formyl group of the f^6^A nucleobase tending to be deprotonated rather than protonated. This detection weakness under positive mode ionization increases the chance of f^6^A being misidentified as m^6,6^A in the absence of synthetic standards. Therefore, it is recommended to use a PFP or ODS column to detect m^6,6^A and f^6^A because of the large difference in retention times under these conditions (Figure 7C).

### 3.3. Uridine and Cytidine Derivatives

The *m*/*z* differences between protonated uridine and cytidine (*m*/*z* 245.1 and 244.1), and between their nucleobases (*m*/*z* 113 and 112) are approximately one. Thus, monitoring of uridine derivatives can be subject to inferring signals from isotopes of cytidine derivatives. For example, the transition *m*/*z* 259.1→127 can detect m^5^U (5-methyluridine), m^3^U (3-methyluridine), and isotopes of m^5^C (5-methylcytidine), m^4^C (*N*^4^-methylcytidine), and m^3^C (3-methylcytidine). Type I, II, and III misidentifications are possible, in some cases, in a single chromatogram monitoring uridine derivatives. Two standard mixtures were prepared to illustrate this complexity: firstly, 1 pmol each of U (uridine), C (cytidine), and Y (pseudouridine); and secondly, 1 pmol each of s^2^U (2-thiouridine), s^4^U (4-thiouridine), ho^5^U (5-hydroxyuridine), m^5^D (5-methyldihydrouridine), and s^2^C (2-thiocytidine). Figure 8A illustrates the monitoring of uridine (*m*/*z* 245.1→113) with isotopic crosstalk from cytidine. However, although the mass of pseudouridine is identical to uridine, the protonated nucleobase of pseudouridine is not seen. Instead, pseudouridine is uniquely identified by the transitions *m*/*z* 245.1→209/179/155. The CID fragmentation of pseudouridine is shown in Figure 8B [34]. The product ions of pseudouridine derivatives, such as m^1^Y (1-methylpseudouridine) and Ym (2’-*O*-methylpseudouridine), can be predicted using this fragmentation pattern to avoid interference with uridine derivatives.

Type I, II, and III potential misidentifications can be observed simultaneously in the mixture of s^2^U, s^4^U, ho^5^U, m^5^D, and s^2^C standards. Figure 8C shows how peaks for all five modified nucleosides appear in one MRM channel (*m*/*z* 261.1→129): s^2^U and s^4^U are structural isomers (type I); s^2^U, s^4^U, ho^5^U, and m^5^D have similar precursor and product ion masses (<0.5 Da difference) (type II); and a cross-talking isotopologue of s^2^C can be observed in the primary s^2^U/s^4^U/ho^5^U/m^5^D channel (type III).

### 3.4. Guanosine Derivatives

Obstacles to the identification of guanosine derivatives center on the possible isomers, particularly among the monomethylated (m^1^G, m^2^G, m^7^G), dimethylated (m^2,2^G, m^2,7^G), and trimethylated (m^2,2^Gm, m^2,7^Gm) guanosines. Figure 9 shows the elution sequence of m^1^G, m^2^G, and m^7^G on various columns, from which the elution sequence of the dimethylated and trimethylated derivatives can be extrapolated. Figure 9 also illustrates the importance of analytical column choice, with only HILIC being capable of resolving the monomethylated guanosines.

## 4. Conclusions

Monitoring the fragmentation of protonated nucleosides into their respective nucleobases using the MRM mode of TQ-MS is useful for detecting RNA modifications. However, it is not practical to synthesize standards for many of these modified nucleosides and, consequently, three types of misidentification can result. Firstly, structural isomers can appear in the same MRM channel and may be closely eluted under HPLC (type I misidentification). Secondly, mass-analogs differing by less than 0.5 Da might also appear in the same channel and may be confused with other analytes (type II). Finally, analytes with mass differences of one or two Da can cause isotopic crosstalk (type III). This study applied a long HPLC runtime of 50 min but some nucleosides could still not be clearly resolved on-column. Therefore, the authors suggest that reports of rapid LC-MS methods for the detection of nucleosides (typically with 5–10 min runtimes) should be examined to rule out the possibility of analyte misidentifications arising from overlapping peaks. Stable isotope labeling of nucleosides can help to prevent type II and III misidentifications, as the mass shifts due to 13C or 15N provide additional molecular composition information [35,36,37]. Potential misidentifications of frequently modified nucleosides are listed in Table 1, Table 2 and Table 3.

Modified nucleosides can contain hydrophobic nucleobases (e.g., those with isopentenyl moieties) and hydrophilic nucleobases (e.g., those with hydroxy moieties), making reversed-phase and HILIC methods suitable for the separation of different groups of nucleosides. The PFP column (Discovery HS F5) separated uridine and cytidine derivatives effectively, and the HILIC column (Acquity BEH amide) was helpful for adenosine and guanosine derivatives. The chromatographic column should be carefully chosen and the HPLC method optimized depending on the target analyte(s). The analytical sensitivity of the nucleosides under positive mode LC-TQ-MS conditions is usually A>G>C>U (uridine being difficult to protonate under positive mode ionization due to its low p*K*a value). Negative mode ionization, derivatization [38], and the careful selection of productions should be considered to improve sensitivity. The degradation, oxidation [39], and spontaneous chemical derivatization [40] of nucleosides during pre-treatment procedures should also be taken into account. We recommend the establishment of strict standards for nucleoside analysis.

## Figures and Tables

**Figure 1 genes-13-00878-f001:**
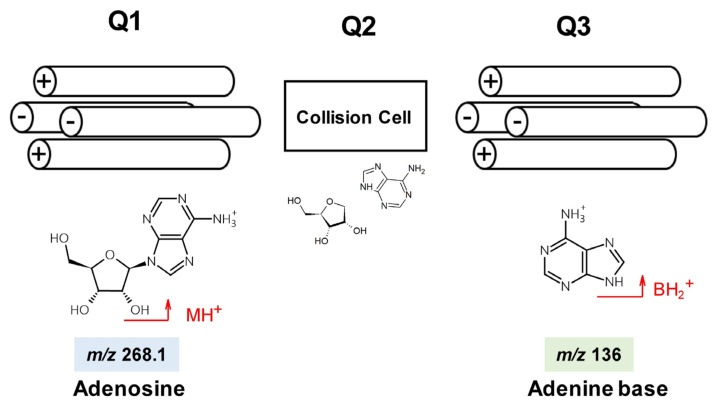
Precursor (MH^+^) and product (BH_2_^+^) ions of adenosine selected by quadrupoles Q1 and Q3.

**Figure 2 genes-13-00878-f002:**
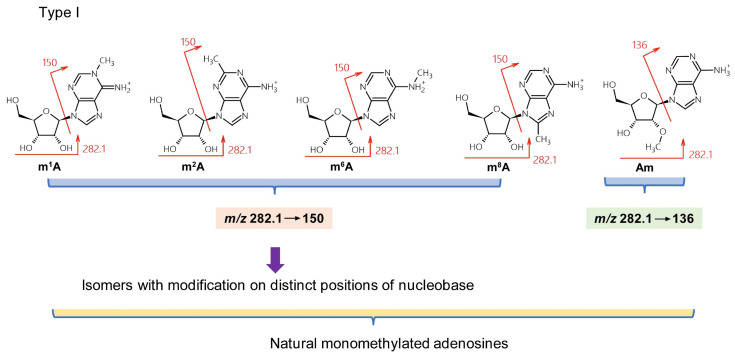
Type I misidentification: natural isomers.

**Figure 3 genes-13-00878-f003:**
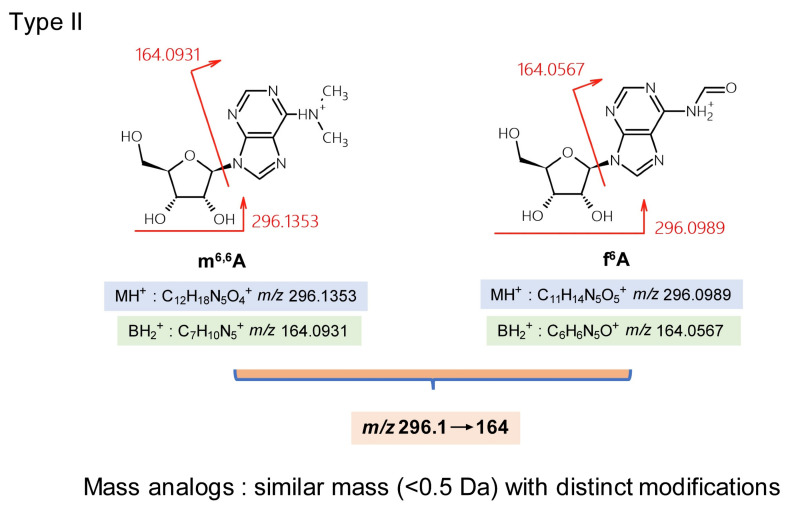
Type II misidentification: mass-analogs (<0.5 Da mass difference).

**Figure 4 genes-13-00878-f004:**
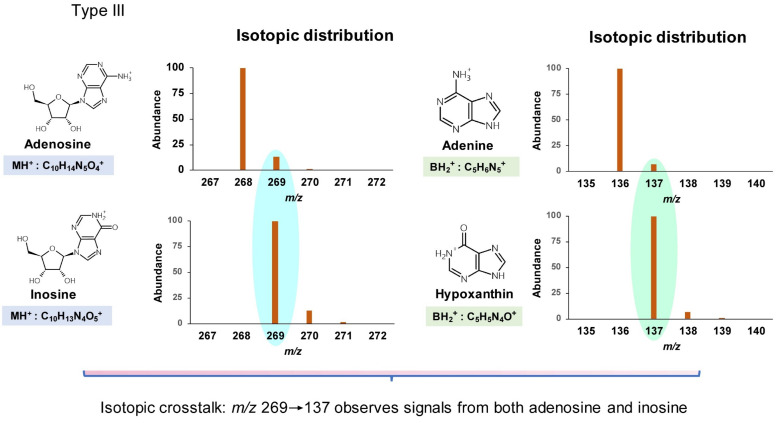
Type III misidentification: isotopic crosstalk. Isotopic distribution was calculated using the online tool https://www.sisweb.com/mstools/isotope.htm (accessed on 1 March 2022).

**Figure 5 genes-13-00878-f005:**
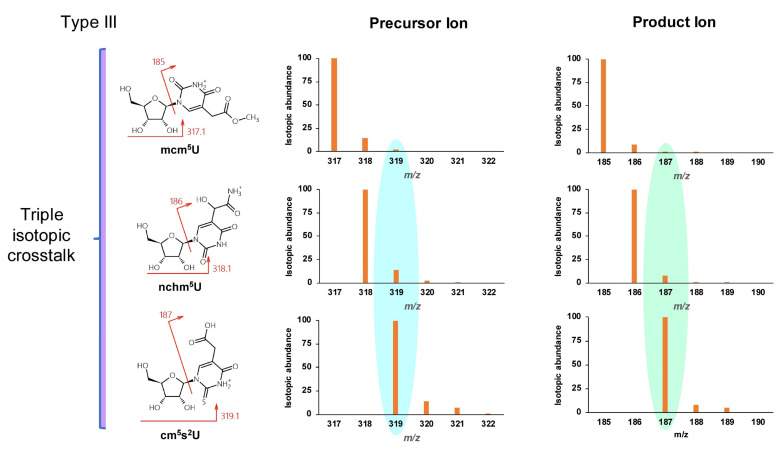
Example of triple isotopic crosstalk type III misidentification between mcm^5^U, mchm^5^U, and cm^5^s^2^U.

**Figure 6 genes-13-00878-f006:**
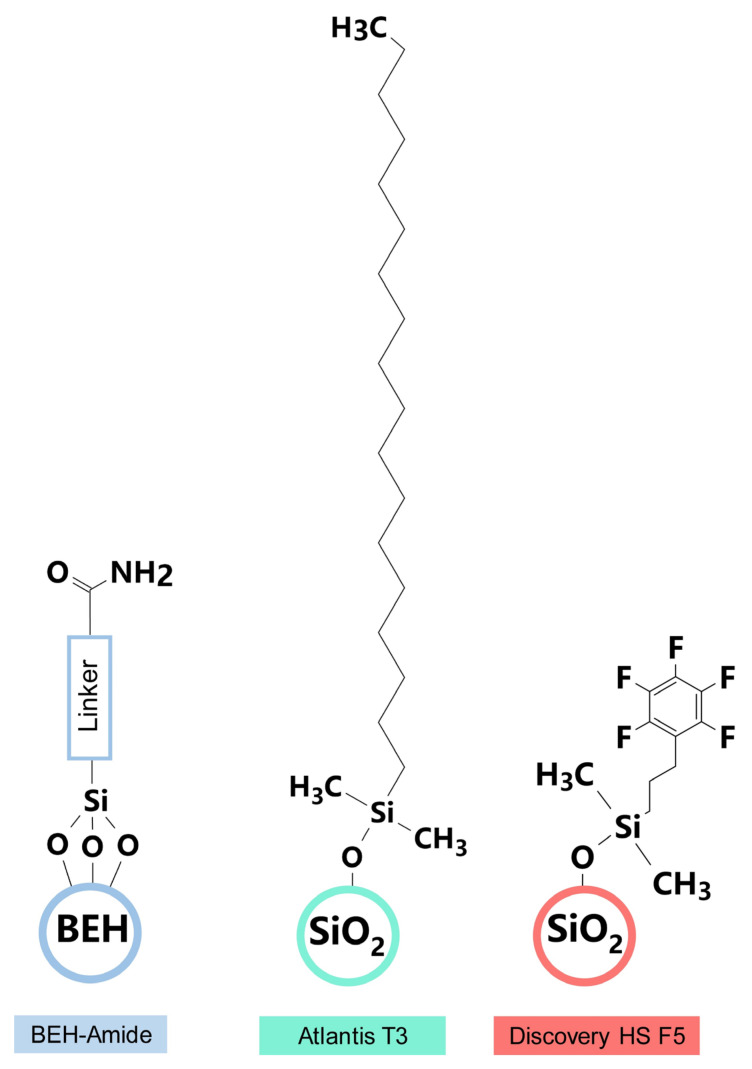
Packing materials and stationary phases of Acquity BEH amide, Discovery HS F5, and Atlantis T3 analytical chromatography columns.

**Figure 7 genes-13-00878-f007:**
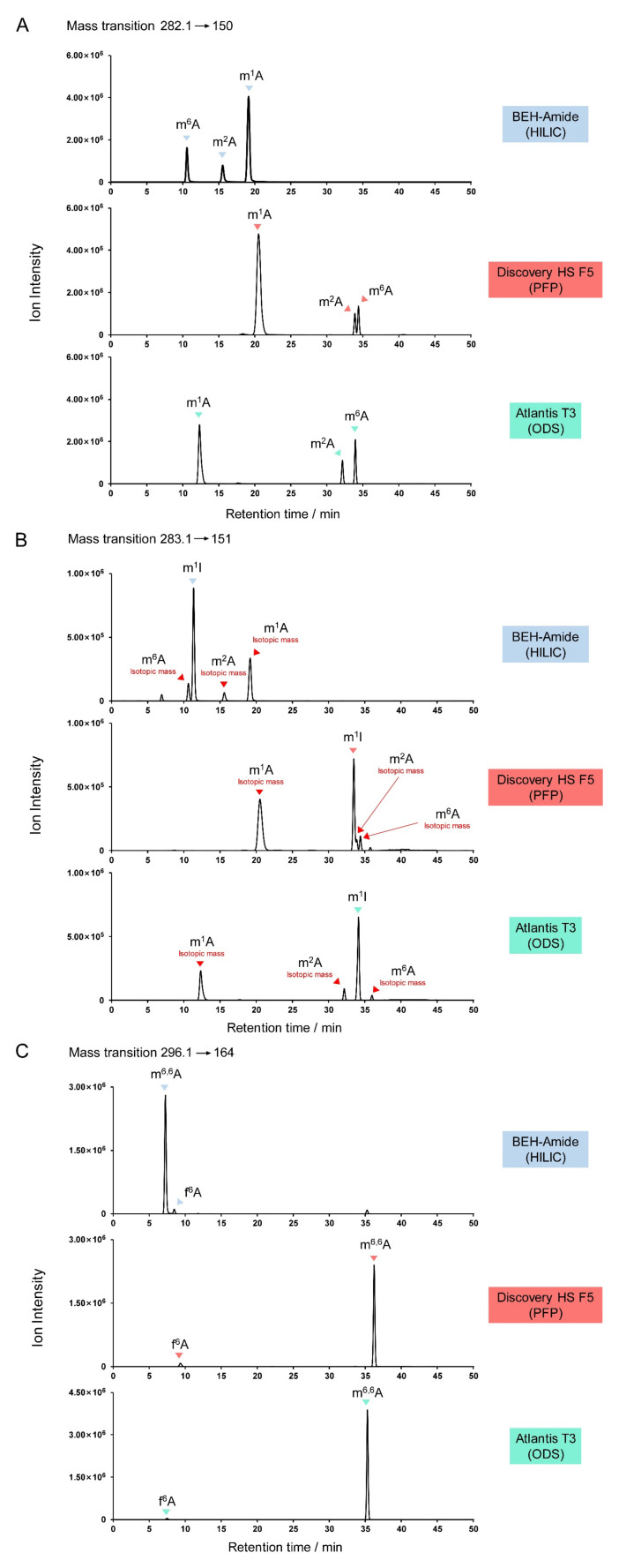
Potential misidentification of adenosine derivatives. (**A**) Separation of natural monomethylated adenosine isomers m^1^A, m^2^A, and m^6^A on various columns. (**B**) Monitoring of 1-methylinosine (m^1^I; *m*/*z* 283.1→151) showing crosstalk from isotopes of m^1^A, m^2^A, and m^6^A. (**C**) Choice of chromatographic column is influenced by the similarity of m^6,6^A and f^6^A retention times in their shared MRM channel (*m*/*z* 296.1→164).

**Figure 8 genes-13-00878-f008:**
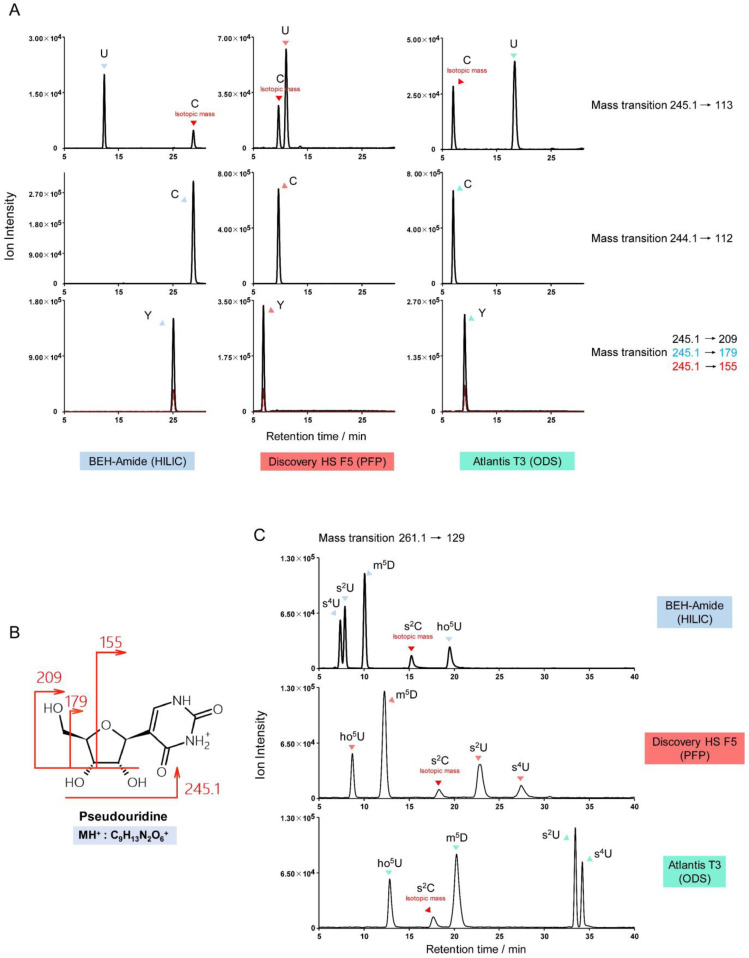
Potential misidentification of cytidine and uridine derivatives. (**A**) Separation of cytidine (labelled C), uridine (labelled U), and pseudouridine (labelled Y) on various columns, showing cytidine crosstalk. (**B**) The unique fragmentation pattern of pseudouridine can be used to predict the product ions of pseudouridine derivatives. (**C**) The MRM channel *m*/*z* 261.1→129 detects the signal from s^2^U, s^4^U, ho^5^U, m^5^D, and isotopic s^2^C, and illustrates three types of potential misidentification.

**Figure 9 genes-13-00878-f009:**
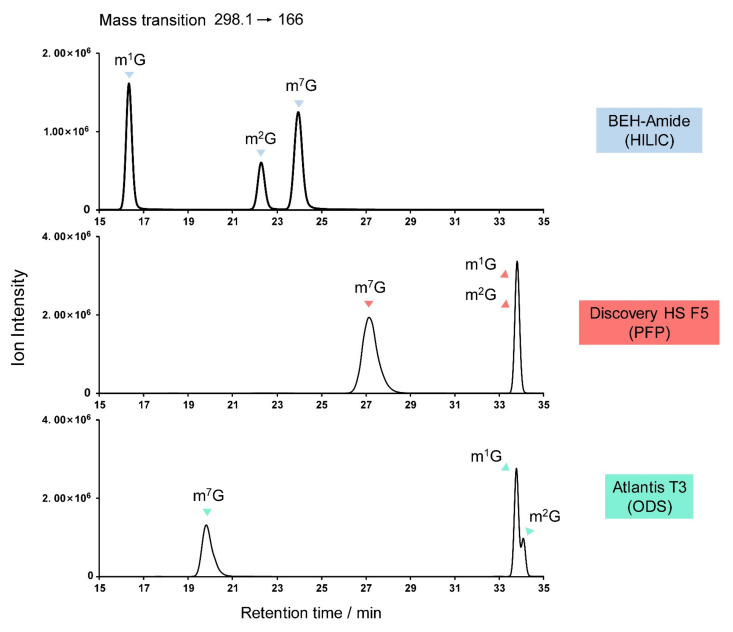
Potential misidentification of guanosine derivatives, showing the separation of the natural monomethylated isomers m^1^G, m^2^G, and m^7^G on various columns.

**Table 1 genes-13-00878-t001:** Nucleosides with potential type I misidentification (natural isomers).

Name	Abbreviation	Chemical Formula	Precursor Ion	Product Ion
**Adenosine derivatives**				
1-Methyladenosine	m^1^A	C_11_H_15_N_5_O_4_	282.1	150
2-Methyladenosine	m^2^A	C_11_H_15_N_5_O_4_	282.1	150
*N*^6^-Methyladenosine	m^6^A	C_11_H_15_N_5_O_4_	282.1	150
8-Methyladenosine	m^8^A	C_11_H_15_N_5_O_4_	282.1	150
2,8-Dimethyladenosine	m^2,8^A	C_12_H_17_N_5_O_4_	296.1	164
*N*^6^,*N*^6^-Dimethyladenosine	m^6,6^A	C_12_H_17_N_5_O_4_	296.1	164
**Cytidine and Uridine derivatives**				
3-Methylcytidine	m^3^C	C_10_H_15_N_3_O_5_	258.1	126
*N*^4^-Methylcytidine	m^4^C	C_10_H_15_N_3_O_5_	258.1	126
5-Methylcytidine	m^5^C	C_10_H_15_N_3_O_5_	258.1	126
3-Methyluridine	m^3^U	C_10_H_14_N_2_O_6_	259.1	127
5-Methyluridine	m^5^U	C_10_H_14_N_2_O_6_	259.1	127
1-Methylpseudouridine	m^1^Y	C_10_H_14_N_2_O_6_	259.1	179
3-Methylpseudouridine	m^3^Y	C_10_H_14_N_2_O_6_	259.1	179
2-Thiouridine	s^2^U	C_9_H_12_N_2_O_5_S	261.1	129
4-Thiouridine	s^4^U	C_9_H_12_N_2_O_5_S	261.1	129
5,2′-*O*-Dimethylcytidine	m^5^Cm	C_11_H_17_N_3_O_5_	272.1	126
*N*^4,2′^-*O*-Dimethylcytidine	m^4^Cm	C_11_H_17_N_3_O_5_	272.1	126
3,2′-*O*-Dimethyluridine	m^3^Um	C_11_H_16_N_2_O_6_	273.1	127
5,2′-*O*-Dimethyluridine	m^5^Um	C_11_H_16_N_2_O_6_	273.1	127
5-Carboxyhydroxymethyluridine	chm^5^U	C_11_H_14_N_2_O_9_	319.1	187
Uridine 5-oxyacetic acid	cmo^5^U	C_11_H_14_N_2_O_9_	319.1	187
5-(Carboxyhydroxymethyl)uridine methyl ester	mchm^5^U	C_12_H_16_N_2_O_9_	333.1	297
Uridine 5-oxyacetic acid methyl ester	mcmo^5^U	C_12_H_16_N_2_O_9_	333.1	297
3-(3-amino-3-carboxypropyl)pseudouridine	acp3Y	C_13_H_19_N_3_O_8_	346.1	214
3-(3-amino-3-carboxypropyl)uridine	acp3U	C_13_H_19_N_3_O_8_	346.1	214
**Guanosine derivatives**				
1-Methylguanosine	m^1^G	C_11_H_15_N_5_O_5_	298.1	166
*N*^2^-Methylguanosine	m^2^G	C_11_H_15_N_5_O_5_	298.1	166
7-methylguanosine	m^7^G	C_11_H_15_N_5_O_5_	298.1	166
1,2′-*O*-Dimethylguanosine	m^1^Gm	C_12_H_17_N_5_O_5_	312.1	166
*N*^2,2′^-*O*-Dimethylguanosine	m^2^Gm	C_12_H_17_N_5_O_5_	312.1	166
*N*^2,7^-Dimethylguanosine	m^2,7^G	C_12_H_17_N_5_O_5_	312.1	180
*N*^2^,*N*^2^-Dimethylguanosine	m^2,2^G	C_12_H_17_N_5_O_5_	312.1	180
*N*^2^,*N*^2,2′^-*O*-Trimethylguanosine	m^2,2^Gm	C_13_H_19_N_5_O_5_	326.1	180
*N*^2,7,2′^-*O*-Trimethylguanosine	m^2,7^Gm	C_13_H_19_N_5_O_5_	326.1	180
Isowyosine	imG2	C_14_H_17_N_5_O_5_	336.1	204
Wyosine	imG	C_14_H_17_N_5_O_5_	336.1	204

**Table 2 genes-13-00878-t002:** Nucleosides with potential type II misidentification (mass-analogs).

Name	Abbreviation	Chemical Formula	Precursor Ion	Product Ion
**Adenosine derivatives**				
*N*^6^-Formyladenosine	f^6^A	C_11_H_13_N_5_O_5_	269.1	164
2,8-Dimethyladenosine	m^2,8^A	C_12_H_17_N_5_O_4_	269.1	164
*N*^6^,*N*^6^-Dimethyladenosine	m^6,6^A	C_12_H_17_N_5_O_4_	269.1	164
*N*^6^-Methyl-*N*^6^-threonylcarbamoyladenosine	m^6^t^6^A	C_15_H_20_N_6_O_8_	427.2	295
*N*^6^-Hydroxynorvalylcarbamoyladenosine	hn^6^A	C_16_H_22_N_6_O_8_	427.2	295
**Cytidine and Uridine derivatives**				
2-Thiocytidine	s^2^C	C_9_H_13_N_3_O_4_S	260.1	128
5-Hydroxycytidine	ho^5^C	C_9_H_13_N_3_O_6_	260.1	128
2-Thiouridine	s^2^U	C_9_H_12_N_2_O_5_S	261.1	129
4-Thiouridine	s^4^U	C_9_H_12_N_2_O_5_S	261.1	129
5-Hydroxyuridine	ho^5^U	C_9_H_12_N_2_O_7_	261.1	129
5-Methyldihydrouridine	m^5^D	C_10_H_16_N_2_O_6_	261.1	129
5-Formylcytidine	f^5^C	C_10_H_13_N_3_O_6_	272.1	140
*N*^4^,*N*^4^-Dimethylcytidine	m^4,4^C	C_11_H_17_N_3_O_5_	272.1	140
5-Methyl-2-thiouridine	m^5^s^2^U	C_10_H_14_N_2_O_5_S	275.1	143
5-Methoxyuridine	mo^5^U	C_10_H_14_N_2_O_7_	275.1	143
5-Formyl-2′-*O*-methylcytidine	f^5^Cm	C_11_H_15_N_3_O_6_	286.1	140
*N*^4^,*N*^4^,2′-*O*-Trimethylcytidine	m^4,4^Cm	C_12_H_19_N_3_O_5_	286.1	140
5-Carbamoylmethyl-2-thiouridine	ncm^5^s^2^U	C_11_H_15_N_3_O_6_S	318.1	186
5-Carbamoylhydroxymethyluridine	nchm^5^U	C_11_H_15_N_3_O_8_	318.1	186
5-Carboxymethyl-2-thiouridine	cm^5^s^2^U	C_11_H_14_N_2_O_7_S	319.1	187
5-Carboxyhydroxymethyluridine	chm^5^U	C_11_H_14_N_2_O_9_	319.1	187
Uridine 5-oxyacetic acid	cmo^5^U	C_11_H_14_N_2_O_9_	319.1	187
5-Methoxycarbonylmethyl-2-thiouridine	mcm^5^s^2^U	C_12_H_16_N_2_O_7_S	333.1	201
5-(carboxyhydroxymethyl)uridine methyl ester	mchm^5^U	C_12_H_16_N_2_O_9_	333.1	201
Uridine 5-oxyacetic acid methyl ester	mcmo^5^U	C_12_H_16_N_2_O_9_	333.1	201
5-Carboxymethylaminomethyl-2-thiouridine	cmnm^5^s^2^U	C_12_H_17_N_3_O_7_S	348.1	216
3-(3-amino-3-carboxypropyl)-5,6-dihydrouridine	acp^3^D	C_13_H_21_N_3_O_8_	348.1	216

**Table 3 genes-13-00878-t003:** Nucleosides with potential type III misidentification (isotopic crosstalk).

Name	Abbreviation	Chemical Formula	Precursor Ion	Product Ion
**Adenosine derivatives**				
1-Methyladenosine	m^1^A	C_11_H_15_N_5_O_4_	282.1	150
2-Methyladenosine	m^2^A	C_11_H_15_N_5_O_4_	282.1	150
*N*^6^-Methyladenosine	m^6^A	C_11_H_15_N_5_O_4_	282.1	150
1-Methylinosine	m^1^I	C_11_H_14_N_4_O_5_	283.1	151
2′-*O*-Methyladenosine	Am	C_11_H_15_N_5_O_4_	282.1	136
2′-*O*-Methylinosine	Im	C_11_H_14_N_4_O_5_	283.1	137
1,2′-*O*-Dimethyladenosine	m^1^Am	C_12_H_17_N_5_O_4_	296.1	150
1,2′-*O*-Dimethylinosine	m^1^Im	C_12_H_16_N_4_O_5_	297.1	151
**Cytidine and Uridine derivatives**				
3-Methylcytidine	m^3^C	C_10_H_15_N_3_O_5_	258.1	126
5-Methylcytidine	m^5^C	C_10_H_15_N_3_O_5_	258.1	126
*N*^4^-Methylcytidine	m^4^C	C_10_H_15_N_3_O_5_	258.1	126
3-Methyluridine	m^3^U	C_10_H_14_N_2_O_6_	259.1	127
5-Methyluridine	m^5^U	C_10_H_14_N_2_O_6_	259.1	127
2-Thiocytidine	s^2^C	C_9_H_13_N_3_O_4_S	260.1	128
5-Hydroxycytidine	ho^5^C	C_9_H_13_N_3_O_6_	260.1	128
2-Thiouridine	s^2^U	C_9_H_12_N_2_O_5_S	261.1	129
4-Thiouridine	s^4^U	C_9_H_12_N_2_O_5_S	261.1	129
5-Hydroxyuridine	ho^5^U	C_9_H_12_N_2_O_7_	261.1	129
5-Methyldihydrouridine	m^5^D	C_10_H_16_N_2_O_6_	261.1	129
5,2′-*O*-Dimethylcytidine	m^5^Cm	C_11_H_17_N_3_O_5_	272.1	126
*N*^4,2′^-*O*-Dimethylcytidine	m^4^Cm	C_11_H_17_N_3_O_5_	272.1	126
3,2′-*O*-Dimethyluridine	m^3^Um	C_11_H_16_N_2_O_6_	273.1	127
5,2′-*O*-Dimethyluridine	m^5^Um	C_11_H_16_N_2_O_6_	273.1	127
5-Hydroxymethylcytidine	hm^5^C	C_10_H_15_N_3_O_6_	274.1	142
5-Aminomethyluridine	nm^5^U	C_10_H_15_N_3_O_6_	274.1	142
5-Methyl-2-thiouridine	m^5^s^2^U	C_10_H_14_N_2_O_5_S	275.1	143
5-Methoxyuridine	mo^5^U	C_10_H_14_N_2_O_7_	275.1	143
5-Cyanomethyluridine	cnm^5^U	C_11_H_13_N_3_O_6_	284.1	152
*N*^4^-Acetylcytidine	ac^4^C	C_11_H_15_N_3_O_6_	286.1	154
5-Formyl-2′-*O*-methylcytidine	f^5^Cm	C_11_H_15_N_3_O_6_	286.1	140
*N*^4^,*N*^4^,2′-*O*-Trimethylcytidine	m^4,4^Cm	C_12_H_19_N_3_O_5_	286.1	140
2′-*O*-Methyl-5-hydroxymethylcytidine	hm5Cm	C_11_H_17_N_3_O_6_	288.1	142
5-Methylaminomethyluridine	mnm^5^U	C_11_H_17_N_3_O_6_	288.1	156
5-Aminomethyl-2-thiouridine	nm^5^s^2^U	C_10_H_15_N_3_O_5_S	290.1	158
5-Carbamoylmethyluridine	ncm^5^U	C_11_H_15_N_3_O_7_	302.1	170
5-Carboxymethyluridine	cm^5^U	C_11_H_14_N_2_O_8_	303.1	171
5-Methylaminomethyl-2-thiouridine	mnm^5^s^2^U	C_11_H_17_N_3_O_5_S	304.1	172
5-Methoxycarbonylmethyluridine	mcm^5^U	C_12_H_16_N_2_O_8_	317.1	185
5-Carbamoylmethyl-2-thiouridine	ncm^5^s^2^U	C_11_H_15_N_3_O_6_S	318.1	186
5-Carbamoylhydroxymethyluridine	nchm^5^U	C_11_H_15_N_3_O_8_	318.1	186
5-Carboxymethyl-2-thiouridine	cm^5^s^2^U	C_11_H_14_N_2_O_7_S	319.1	187
5-Carboxyhydroxymethyluridine	chm^5^U	C_11_H_14_N_2_O_9_	319.1	187
Uridine 5-oxyacetic acid	cmo^5^U	C_11_H_14_N_2_O_9_	319.1	187
5-Carboxymethylaminomethyluridine	cmnm^5^U	C_12_H_17_N_3_O_8_	332.1	200
5-Methoxycarbonylmethyl-2-thiouridine	mcm^5^s^2^U	C_12_H_16_N_2_O_7_S	333.1	201
5-(carboxyhydroxymethyl)uridine methyl ester	mchm^5^U	C_12_H_16_N_2_O_9_	333.1	201
Uridine 5-oxyacetic acid methyl ester	mcmo^5^U	C_12_H_16_N_2_O_9_	333.1	201
5-Carboxymethylaminomethyl-2′-*O*-methylu-ridine	cmnm^5^Um	C_13_H_19_N_3_O_8_	346.1	200
5-(carboxyhydroxymethyl)-2′-*O*-methyluridi-ne methyl ester	mchm^5^Um	C_13_H_18_N_2_O_9_	347.1	201
2′-*O*-Methyluridine 5-oxyacetic acid methyl ester	mcmo^5^Um	C_13_H_18_N_2_O_9_	347.1	201
5-(isopentenylaminomethyl)-2-thiouridine	inm^5^s^2^U	C_15_H_23_N_3_O_5_S	358.1	226
1-Methyl-3-(3-amino-3-carboxypropyl)pseud-ouridine	m^1^acp3Y	C_14_H_21_N_3_O_8_	360.1	228
**Guanosine derivatives**				
7-Aminocarboxypropylwyosine methyl ester	yW-58	C_19_H_26_N_6_O_7_	451.2	319
Undermodified hydroxywybutosine	OHyWx	C_18_H_24_N_6_O_8_	453.2	321

## Data Availability

The data supporting this study’s findings are available on http://www.doi.org/10.11922/sciencedb.01462 (accessed on 1 March 2022) with appropriate protection periods.

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
