# Peer review of "Potential Misidentification of Natural Isomers and Mass-Analogs of Modified Nucleosides by Liquid Chromatography–Triple Quadrupole Mass Spectrometry"

_genes, 2022, doi:10.3390/genes13050878_

Round 1
Reviewer 1 Report
The paper “Potential misidentification of natural isomers and mass-analogs of modified nucleosides by liquid chromatography-triple quadrupole mass spectrometry” by Lin et al. describes the analysis of 17 nucleoside standards by LC-TQ-MS with separation on three different analytical columns and discusses about its misidentifications.
The topic is extremely important and useful contribution for identification of modified nucleosides by LC-TQ-MS.
I wish the authors address the following comments. I believe that the manuscript can be accepted after minor comments below are addressed.
(1)
In Figure 2, what is the chemical structure of the “Linker” moiety of the packing materials of BEH amide? Please improve the figure in the text or add some comments in your manuscript.
(2)
The resolution of all figures should be improved.
(3)
In all figures, “Fig.1. A” or “Fig. 1B” etc. IN THE FIGURE are not necessary. Instead, use “(A)” or “(B)”.
Reviewer 2 Report
In this work, the authors analyzes the major causes of analyte misidentification in the determination of nucleosides by LC-TQ-MS
INTRODUCTION
- In the Figure 1, the authors need to replace the figure captions inserted in the figure, Fig 1A, Fig 1B,… by A), B),…. Additionally, since a high amount of text is included in the introduction, Figure 1 should be divided into different figures, and these figures should be embedded in the manuscript after each type of misidentification (for a better understanding of the introduction)
-Line 90: The term “structural isomers” seems not to be enough clear to describe this phenomenon. Maybe “regioisomers” could be a more clear alternative?
MATERIALS AND METHODS
- The authors did not include the information of the suppliers…. Howei 151 Pharm (country), Aladdin (country), among others
RESULTS AND DISCUSSION
- The same than for figure 1 ……A), B),….
- Despite the results are interesting, I do not observe a detailed discussion and explanation of the results.
